# Photon pumping, photodissociation and dissipation at interplay for the fluorescence of a molecule in a cavity

**Megha Gopalakrishna[1], Emil Viñas Boström[2] and Claudio Verdozzi[3]⋆**

**1** Department of Physics, Division of Mathematical Physics, Lund University,
22100 Lund, Sweden
**2** Max Planck Institute for the Structure and Dynamics of Matter,
Luruper Chaussee 149, 22761 Hamburg, Germany
**3** Department of Physics, Division of Mathematical Physics and ETSF,
Lund University, 22100 Lund, Sweden

⋆ Claudio.Verdozzi@teorfys.lu.se

## Abstract

We introduce a model description of a diatomic molecule in an optical cavity, with pump and fluorescent fields, and electron and nuclear motion are treated on equal footing and exactly. The model accounts for several optical response temporal scenarios: A Mollow spectrum hindered by electron correlations, a competition of harmonic generation and molecular dissociation, a dependence of fluorescence on photon pumping rate and dissipation. It is thus a general and flexible template for insight into experiments where quantum photon confinement, leakage, nuclear motion and electronic correlations are at interplay.



# 1  Introduction

Second harmonic generation (SHG) is the conversion by some material system of two photons of frequency $\omega$ into a single photon of frequency $2\omega$. A classic hallmark of nonlinear optical behavior [1], SHG still is, sixty years after its discovery [2], the focus of extensive research in physics [3], engineering [4], chemistry [5], biology [6], and medicine [7]. Part of this interest stems from technology [8,9]: SHG is the operating mechanism in optical devices and imaging techniques that are surface or interface sensitive [10–12]. Another reason is that there are aspects and regimes of SHG still not fully understood, making it a valuable benchmark for advances in nonlinear optics.

Several theoretical methods are used to describe SHG [13], from nonlinear response in frequency space [14] to Bloch-Maxwell equations [15] and real-time first-principle approaches [13, 16–19]. Often, classical radiation fields are used, which is appropriate in the strong field limit. However, highly interesting effects in SHG (and fluorescence in general) appear in the low photon regime [20–23], where quantum effects generally dominate [24] and the so-called rotating wave approximation (RWA) [25–29] may be inadequate [30–32].

Optical cavities permit an accurate selection of confined electromagnetic modes [33–36], and allow to address the low photon regime of SHG [37]. However, a key element left out of many theoretical works on few-level systems is an explicit description of electronic correlations and nuclear dynamics, even though these can importantly affect the harmonic signal [38–40]. First-principle descriptions include these contributions [13,14,16,19], but usually approximations are made in numerical implementations. Therefore, because of the broad relevance of SHG, it is useful to consider model systems where photon pumping, cavity leakage, electronic correlations, and nuclear motion can be treated exactly and on equal footing, to gain a generic and accurate understanding of their interplay.

In this work we introduce a simple and flexible theoretical framework to describe a single molecule embedded in an optical cavity, and study its fluorescence properties. Within this framework all the aforementioned effects and interactions are considered, and the following picture emerges: (1) the SHG signal is larger for faster photon pumping; (2) electron-electron interactions strongly reduce the fluorescence signal; (3) for light atomic masses photodissociation takes place, inhibiting fluorescence and SHG; for heavier masses, the opposite occurs; (4) both resonant and SHG signals are quenched in time by cavity leakage. While not tied to any specific molecule, our results unveil a multifaceted light-matter scenario for SHG and fluorescence in the low photon regime, when multi-photon effects are important. At the same time, they give qualitative but rigorous initial insight for more refined investigations of systems of direct experimental interest.

## 2 Hamiltonian, initial state and fluorescent spectrum

We consider a homonuclear diatomic molecule embedded in a cavity, where each atom has a mass $M$ and a single $s$-orbital. The molecule is occupied by two electrons of opposite spin, interacting with a cavity field of frequency $\omega_0$ and a fluorescent field of frequency $\omega$. The molecule and cavity are assumed to be one-dimensional, with the molecular axis aligned with the axis of the cavity. The total Hamiltonian reads

$$\hat{H}(t) = \hat{H}_s(t) + \hat{V}_{\text{ext}}(t), \tag{1}$$

$$\hat{H}_s(t) = \hat{H}_{\text{mol}} + \hat{H}_{\text{rad}} + \hat{H}_{\text{int}}(t), \tag{2}$$

where $\hat{H}_{\text{mol}}$, $\hat{H}_{\text{rad}}$ and $\hat{H}_{\text{int}}(t)$ respectively describe the molecule, the photon fields, and the light-matter interaction [33]. The external field term, $\hat{V}_{\text{ext}}(t)$, will be discussed at the end of this section. In more detail, the molecular Hamiltonian we use is

$$H_{\text{mol}} = \frac{\hat{P}^2}{2(2M)} + \frac{\hat{p}^2}{2(M/2)} + \frac{C}{\hat{x}^4} + U \sum_i \hat{n}_{i\uparrow}\hat{n}_{i\downarrow} - V e^{-\lambda \hat{x}} \sum_\sigma \left( c_{1\sigma}^\dagger c_{2\sigma} + c_{2\sigma}^\dagger c_{1\sigma} \right), \tag{3}$$

where the first two terms give the kinetic energy of the molecular center of mass (with momentum $\hat{P}$), and relative atomic motion (with momentum $\vec{p}$). The third term accounts for an inter-atomic repulsion of strength $C$, with $\hat{x}$ the inter-atomic coordinate. The fourth term represents an intra-orbital repulsive interaction of strength $U$ between the electrons, where $\hat{n}_{i\sigma} = c_{i\sigma}^\dagger c_{i\sigma}$ and $c_{i\sigma}^\dagger$ creates an electron with spin projection $\sigma$ at atom $i$.

Finally, the last term in $\hat{H}_{\text{mol}}$ describes the electron kinetic energy arising from electrons hopping between the atoms. The strength of this contribution is proportional to $V$, but it also depends on the internuclear distance via the operator $e^{-\lambda \hat{x}}$ (with $\lambda$ an attenuation parameter). This gives a phenomenological (but intuitively physically plausible [41–44]) fully quantum mechanical interaction between the electrons and the inter-atomic motion. In the numerical calculations, we set $V = 2$, $C = 0.6$ and $\lambda = 0.6$, to obtain a Morse-like potential landscape for inter-atomic motion, and an equilibrium position $r_0 = 1.156$. In this way, the effective hopping $V_{\text{eff}} = V \exp(-\lambda r_0) = 1$ within few parts per thousand.

The second contribution to $\hat{H}_s$ describes the two photon modes, $\hat{H}_{\text{rad}} = \omega_0 b^\dagger b + \omega b'^\dagger b'$, with $b$ ($b'$) destroying a cavity (fluorescent) photon with frequency $\omega_0$ ($\omega$). For computational simplicity we exclude the direct interaction between modes and nuclei, and neglect center of mass motion.[1] The cavity-molecule interaction is thus $\hat{H}_{\text{int}} = \hat{\mathcal{M}}\left[ g_c(b^\dagger + b) + g'(t)(b'^\dagger + b') \right]$, where $\hat{\mathcal{M}} = \sum_\sigma (c_{b\sigma}^\dagger c_{a\sigma} + c_{a\sigma}^\dagger c_{b\sigma})$ and $c_{b/a} = (c_1 \pm c_2)/\sqrt{2}$ destroys an electron in the molecule's bonding or antibonding state. In the calculations, the fluorescent coupling is damped, i.e. $g'(t) = g_f \exp(-\Gamma t)$ (we set $\Gamma = 0.02$), to describe phenomenologically cavity losses [22, 37]. Later in the paper, we will supplement this phenomenological dissipation with a more rigorous description of cavity leakage, by coupling the system to baths of harmonic oscillators.

It useful at this point to briefly comment on these two ways to affect the fluorescence response: The phenomenological damping due to $\Gamma$ acts on the coupling between the matter and fluorescent photons, to account in an effective way for the fact that the spontaneous emission into a photon continuum is described via a single effective mode. On the other hand, with the bath of harmonic oscillators, we describe a dissipation channel for the photon modes, i.e. for the finite cavity quality. Since the photon-photon coupling utilised with the harmonic bath can be seen as an effect of all photon modes interacting via the molecular system, the two effect are clearly related, and yet rather distinct.

---

[1]This is of no consequence for a rigid molecule, but can have a role in general. We are currently developing a semiclassical description of the interaction between photon modes and nuclear charge and include its effect on the motion of the nuclei.

We will consider two initial light+matter states: i) A product state $|\Psi_0'\rangle \equiv |g_m\rangle|\beta\rangle_c|0\rangle_f$, with the molecule in its ground state $|g_m\rangle$ for $g_c = g_f = 0$, the cavity field in a coherent state $|\beta\rangle_c$, and the fluorescence field in its vacuum state $|0\rangle_f$. ii) The ground state $|\Psi_0''\rangle \equiv |g\rangle$ of the full Hamiltonian $\hat{H}_s(t = 0)$.

Lastly, we discuss the external field $\hat{V}_{\text{ext}}(t)$. This represents the action of a laser injecting into the cavity incident photons with frequency $\omega_0$. As specified next, $\hat{V}_{\text{ext}}(t)$ always acts only in the initial part of the simulation interval; in other words, $\hat{H}(t)$ and $\hat{H}_s(t)$ are time-independent at long times. Explicitly, the form chosen is $\hat{V}_{\text{ext}}(t) = g_d(b^\dagger + b)[f(t)\sin\omega_0 t]$, with a) $f(t) = \theta(t_s - t)$ a step envelope vanishing after time $t_s$ or b) $f(t)$ a smoothened rectangular pulse. The rectangular pulse $f(t)$ acts approximately between $t_1$ and $t_2$, with envelope $f(t) = [1 - \mathcal{F}_1(t)]\mathcal{F}_2(t)$, where $\mathcal{F}_i(t) = [\exp((t - t_i)/\tau_i) + 1]^{-1}$. In all calculations, $\tau_1 = \tau_2 = 2.0$ whilst the values of $t_1, t_2$ are case specific, and reported in the figure captions.

## 2.1 Resonance frequency and fluorescence spectrum

We consider a cavity mode with a frequency of either $\omega_0 = \Omega_R$ in resonance with the molecule's electronic transitions, or $\omega_0 = \Omega_R/2$. Due to space and spin symmetries, the molecule's electronic ground state is a spin singlet of even parity. Since the total electron spin $S$ is conserved in absorption and emission, $\Omega_R = E_{\text{odd},S=0}^{\text{ex}} - E_{\text{even},S=0}^g = U/2 + [4V_{\text{eff}}^2 + (U/2)^2]^{1/2}$ (see Appendix A.1).[2] Concerning the value chosen for the interaction among the electrons, in Appendix A.2 we show that fluorescence weakens on increasing the electronic correlations. Accordingly, in the rest of the paper we focus on the weakly interacting regime where $U = 1.0$ and $\Omega_R = 2.56$.

We characterize the fluorescence spectrum in terms of

$$\mathcal{P}(t, \omega) = \sum_{\lambda r_i n} \sum_{m>0} \left| \langle \lambda r_i nm | \mathcal{T}[e^{-i\int_0^t \hat{H}(t')dt'}] | \Psi_0 \rangle \right|^2 , \qquad (4)$$

where $\mathcal{P}$ is the probability to have one or more photons in the fluorescence mode $\omega$ at time $t$ [22]. Here $|\Psi_0\rangle$ is a given initial state (i.e., either $|\Psi_0'\rangle$ or $|\Psi_0''\rangle$ above) and the $\omega$-dependence is contained in $\hat{H}(t)$. The sums over $\lambda$, $r_i$ and $n$ trace out electronic, nuclear and cavity mode degrees of freedom, while the sum over $m$ ensures that at least one fluorescent photon is emitted. The real-time dynamics of the system (with coupled electronic, atomic and photonic degrees of freedom) was obtained via the short iterated Lanczos algorithm, by computing the exact time evolved many-body state $|\Psi(t)\rangle$ starting from $|\Psi_0\rangle$. The configuration size of the problem is $N = 4N_c N_f N_R$, where 4 is the dimension of the electronic subspace, and $N_c$, $N_f$, and $N_R$ are respectively the maximum number of cavity photons, fluorescence photons, and grid points for the nuclear coordinate $x$. We have ensured numerical convergence with respect to these parameters.

## 3 Fluorescence in a rigid molecule and initial state preparation

In a cavity with low photon number, SHG is remarkably sensitive to the system's initial state. This important point is illustrated by comparing the spectra resulting from the different initial states $|\Psi_0'\rangle$ and $|\Psi_0''\rangle$ introduced earlier. With $|\Psi_0'\rangle$, which is a coherent state with $\beta^2$ photons and not an eigenstate of $\hat{H}_s(t)$, the system evolves under the full Hamiltonian $\hat{H}_s(t)$ and $\hat{V}_{\text{ext}} = 0$. Thus, fluorescence photons are emitted in time. For $|\Psi_0''\rangle$, and with the parameters

---

[2]A different prescription could be to consider, irrespective of the value of $U$, an incident frequency in resonance with the one particle levels i.e. $\omega_0 = 2|V_{eff}|$. Within the perspective adopted here, this would simply amount to have an off-resonant incident field, with detuning $\pm|2V_{eff} - \Omega_R|$.

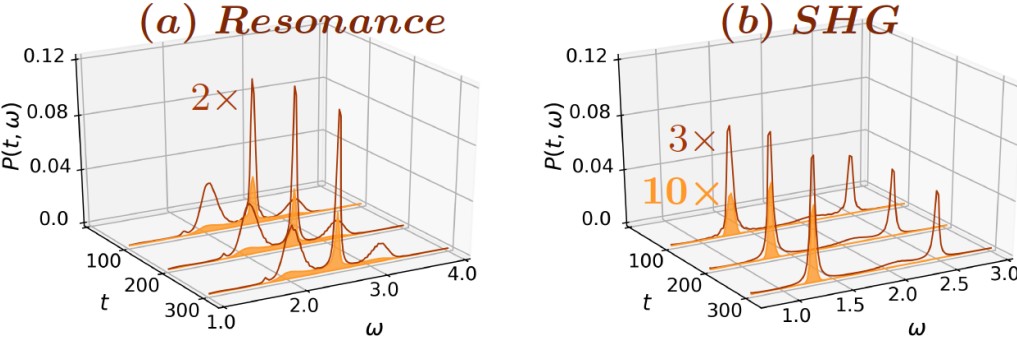

Figure 1: (a) Resonant response for $\omega_0 = \Omega_R$ and (b) SHG response for $\omega_0 = \Omega_R/2$ of a rigid molecule, starting from a coherent state $|\Psi_0'\rangle$ with $\beta^2 = 9$ (empty curves) and from the cavity+molecule's ground state $|\Psi_0''\rangle$ followed by pumping (filled curves). For the pumped cavity, the drive is kept on until $\langle b^\dagger b \rangle \approx 9$. We use $t_1 = \frac{6\pi}{\omega_0}$ and $t_2 = \frac{31\pi}{\omega_0}$, with $g_d = 0.229$ and $0.0996$ in (a) and (b) respectively. In all panels, $U = 1.0$, $g_c = 0.08$, $g_f = 0.01$, $\Omega_R = 2.56$, and $\Gamma = 0.02$. Plots are scaled for visual clarity and the scaling factors are indicated in color.

we consider, the initial occupation of the cavity mode is negligible ($< 10^{-3}$). So, for a meaningful comparison with the results from $|\Psi_0'\rangle$, the cavity is pumped by a driving field $V_{\text{ext.}}(t)$ of frequency $\omega_0$, until an approximately coherent state with average photon number $\langle b^\dagger b \rangle \approx \beta^2$ is reached. The spectra for the two initial configurations, and the low photon limit[3] $\beta = 3$ are in Fig. 1, for both the resonant ($\omega_0 = \Omega_R$) and SHG ($\omega_0 = \Omega_R/2$) cases. In the resonant case, and starting from $|\Psi_0'\rangle$ (Fig. 1a, empty curves), a spectrum with well-defined Mollow features emerges already at early times and converges to a similar profile at longer times. These features can be understood from a dressed-level picture [22, 37] since the cavity mode is in resonance with a parity allowed transition. Interestingly, starting from $|\Psi_0''\rangle$ and pumping the cavity up to $\beta = 3$ (Fig. 1a, filled curves), the spectrum at long times is qualitatively similar to Fig. 1a empty curves, although the intensity of the Mollow sidebands is reduced compared to the main peak. A markedly different picture emerges in the SHG regime: For initial state $|\Psi_0'\rangle$ (Fig. 1b, empty curves), the spectrum quickly develops two sharp features (with a broad shoulder in the middle) corresponding to a Rayleigh (SHG) contribution at $\omega_0$ ($2\omega_0$). However, when starting from the full ground state $|\Psi_0''\rangle$ and pumping the cavity, the SHG signal is strongly suppressed at all times (Fig. 1b, filled curves). In other words, the SHG signal strongly depends on the pumping rate, i.e. on the value of $g_d$.

## 3.1 The dependence on the initial conditions

To uphold our last statement, we consider for simplicity SHG in a two-level system (TLS) with levels $|0\rangle$ and $|1\rangle$ and $\omega_0 = \Omega_R/2$. In Fig. 2a we show the evolution of the total parity $\Pi = \langle e^{i\pi b^\dagger b}(\hat{n}_0 - \hat{n}_1)e^{i\pi b'^\dagger b'}\rangle$, the cavity mode occupation, and the occupation $n_1$ of the TLS excited state. The dynamics is obtained starting either from a product state with the cavity mode in a coherent state (with $\beta^2 = 1$), or from the exact ground state where the cavity mode is pumped at different speeds until $\langle b^\dagger b \rangle \approx 1$.

Fig. 2b shows the corresponding long-time limit SHG. When starting from $|\Psi_0'\rangle$, $\Pi$ has a constant mixed parity $\Pi_{\text{coh}} \approx 0.17$. By contrast, when starting from $|\Psi_0''\rangle$, initially $\Pi$ is 1, but then drops to $\Pi_{\text{coh}}$ with pumping. Thus, in both cases and at almost all times, the system has mixed parity (which is necessary for SHG in a TLS [37]). Yet, the SHG signal is absent for

---

[3] Even with $\beta = 3$, the size of the incident photon subspace $N_i$ must be much larger (explicitly, $N_i = 60$) to have good numerical convergence.

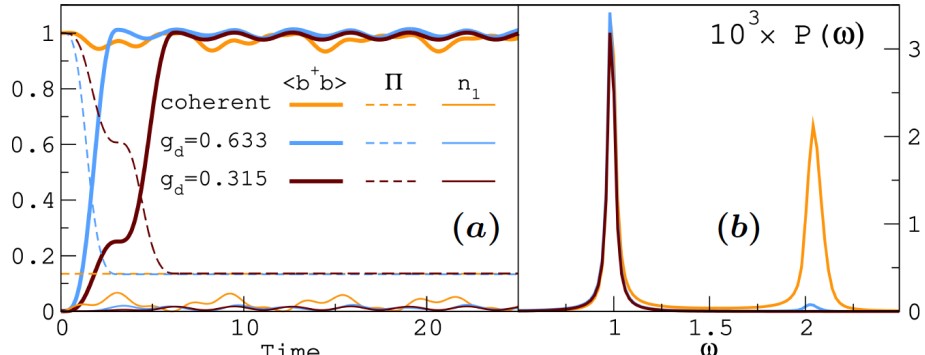

Figure 2: Cavity pumping in a two level system with $g_c = 0.1, g_f = 0.01, \Gamma = 0.02$, and $\omega_0 = \Omega_R/2 = 1$. Starting from the same ground state, two pumping speeds are considered with $t_s = \frac{\pi}{\omega_0}$ and $t_s = \frac{2\pi}{\omega_0}$ respectively. Reference results from an initial coherent state ($\beta^2 = 1$) and no pumping are also shown. (a) Time-evolved average number of cavity photons, total parity and excited state population. (b) Corresponding SHG spectra at long times.

slow pumping and very small for fast ramping. Further insight comes from how the population $n_1$ of the excited level changes in time: it is very small for the pumped cases, but noticeably larger for the coherent case. Thus, the cavity pumping speed strongly affects the population of the excited level and the SHG strength, which increases for faster drives, and similar trends are observed for the resonant regime (see Appendix A.3). While exemplified for a TLS, our considerations equally hold for the molecule investigated in the rest of the paper.

## 4 Cavity leakage and atomic motion

For a more microscopic treatment of the cavity leakage, we now couple both photon modes $\omega_0, \omega$ to two baths of independent classical oscillators (with variables $\{x_k, p_k\}$ and $\{x'_k, p'_k\}$). The couplings of baths and photon modes are of the Caldeira-Leggett type [45–47], and add a contribution $\hat{H}_{\text{leak}}$ to the system's Hamiltonian of Eq. (1), with

$$\hat{H}_{\text{leak}} = \frac{1}{2} \sum_{k=1}^{N_B} \left[ \left( p_k^2 + p_k'^2 \right) + \omega_k^2 \left( x_k^2 + x_k'^2 \right) \right] - \sum_{k=1}^{N_B} C_k \left[ x_k \left( b^\dagger + b \right) + x_k' \left( b'^\dagger + b' \right) \right]. \quad (5)$$

In the presence of the baths, the frequency of the modes gets renormalized via $\omega_0 \to \omega_0 + \sum_{k=1}^{N_B} C_k^2/\omega_k^2$ and $\omega \to \omega + \sum_{l=1}^{N_B} C_l^2/\omega_l^2$. Furthermore, an additional "counterterm" $\hat{V}_{\text{count.}} \propto [(b^\dagger)^2 + b^2 + (b'^\dagger)^2 + b'^2]$ appears in the Hamiltonian (see Appendix A.4 for details), and its role is discussed in Appendix A.5. In the actual calculations, $\omega_k = k\Delta$ and $C_k = A\omega_k^a$. The values of $N_B$, $A$, $\Delta$ and $a$ determine the decay rate of the photons (the cavity quality). The bath variables are propagated via Ehrenfest dynamics. For example, for the $\{x_k, p_k\}$ bath, $\ddot{x}_k(t) = -\omega_k^2 x_k(t) + C_k \langle b^\dagger + b \rangle_{\bar{x}, t}$, where $\bar{x} \equiv \{x_k\}$. In turn, the coordinates $\bar{x}, \bar{x}'$ enter parametrically into the wave function $|\Psi(t)\rangle$ of the quantum subsystem (i.e. the photon modes plus the molecule).

Using the Ehrenfest approximation could introduce a problem with detailed balance. However, since our approach to cavity leakage does not aim to a quantitative realistic description, but rather to explore/illustrate qualitative trends, an incorrect detailed balance is not expected to not be a crucial hampering factor. Furthermore, while computationally inexpensive, this

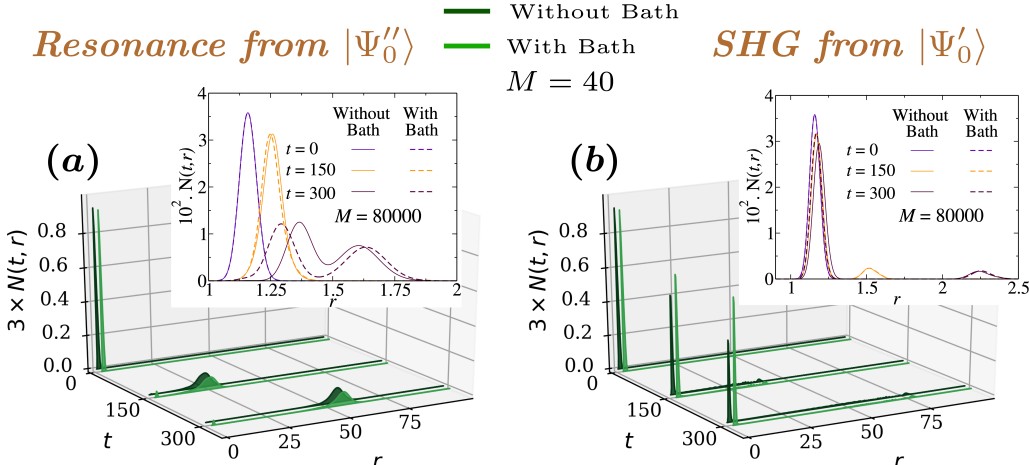

**Figure 3:** Dynamics of the relative interatomic distance in resonant (a) and SHG (b) regimes, for atomic masses $M = 40$ (main plots) and $M = 8 \times 10^4$ (insets). In all panels $U = 1$ and $r_0 = 1.156$. At resonance (a) the calculations were performed by pumping the cavity until $\langle b^\dagger b \rangle \approx 9$ starting from the interacting ground state $|\Psi_0''\rangle$, with $g_c = 0.03$, $g_f = 0.01$ and $g_d = 0.151$. For SHG (b) the calculations started from the product state $|\Psi_0'\rangle$ with the cavity field in a coherent state with $\beta^2 = 9$, with $g_c = 0.08$, $g_f = 0.01$ and $\omega_0 = 1.28$. In all cases displayed, the phenomenological cavity dissipation coefficient $\Gamma = 0.02$, either when the baths are included or not. The values of the bath parameters are the same for the incident and the fluorescent fields. They are $C_k = A(\Delta k)^a$, $N_B = 1000$ oscillators, $A = 0.005$, $a = 0.6$ and $\Delta = 0.01$.

treatment of the bath keeps the quantum dynamics at the many-body wavefunction level unitary and Hermitian.

## 4.1 Nuclear motion

Until now, the molecule was kept rigid at interatomic distance $r_0$ corresponding to the maximum of $N(t = 0, r)$, the equilibrium probability distribution of the nuclear relative coordinate $r$. How the interatomic distance is affected by the light-matter interaction (and viceversa) is shown in Fig. 3, where we display time snapshots of $N(t, r)$ for both resonant and SHG regimes. In these simulations, cavity leakage is included via the oscillator baths, whilst other sources of dissipation are still taken into account via an exponential attenuation ($g'(t) = g_f e^{-\Gamma t}$). In the resonant regime, the system is initially in its ground state $|\Psi_0''\rangle$ and the cavity mode is subsequently pumped. In this case, the molecule dissociates quite rapidly when $M = 40$, irrespective of the presence of the bath. Conversely, for the larger mass, no dissociation occurs in the simulation interval, and the atoms remain around the equilibrium configuration with a broadened distribution $N(t, r)$.

In the SHG regime, the system's initial state is $|\Psi_0'\rangle$ for both values of $M$. Here, the molecule predominantly remains close to the equilibrium configuration at all times, especially when leakage is added. That is, the tendency to delocalise is greater when only the exponential damping is present, indicating that cavity leakage also plays a role. As shown next, the different atomic dynamics affect the optical response in distinct ways.



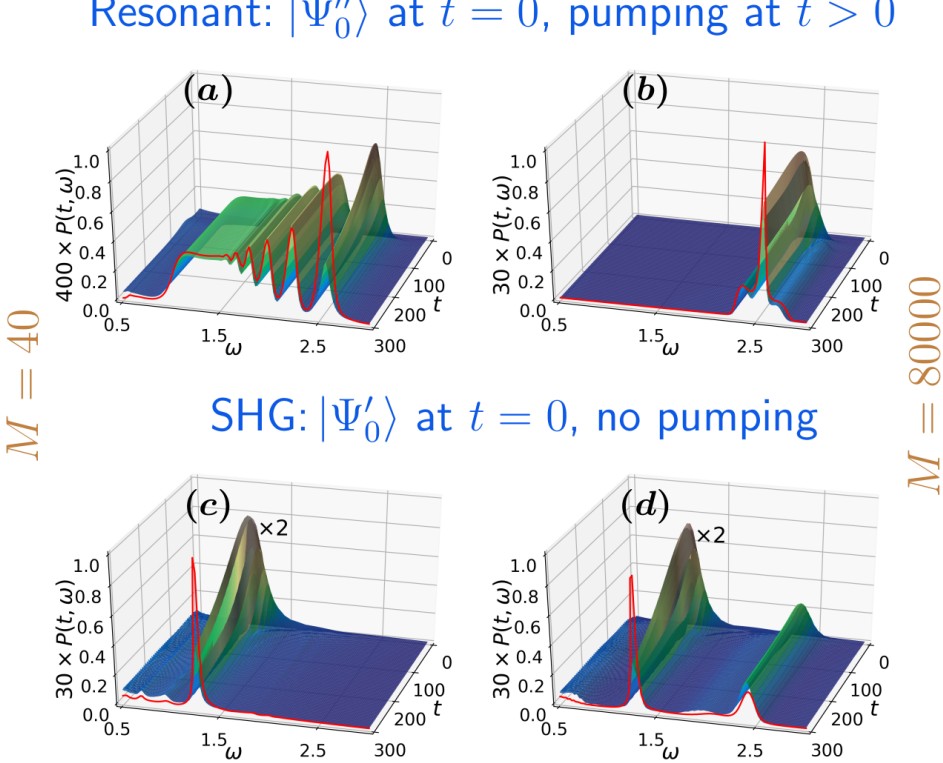

Figure 4: Time-dependent fluorescence for atomic masses $M = 40$ (a ,c) and $M = 8 \times 10^4$ (b, d). The time evolution was performed with the classical baths, whilst the red curves show the long-time limit of $\mathcal{P}(t, \omega)$ in the absence of leakage. The bath parameters are the same for the bath of the incident field and the one of the fluorescent field. They are $N_B = 1000$ oscillators, with $C_k = A(\Delta k)^a$, $A = 0.005$, $a = 0.6$ and $\Delta = 0.01$. In all the calculations $\Gamma = 0.02$. (a,b) Resonant case, starting from $|\Psi_0''\rangle$ and pumping the cavity until and $\langle b^\dagger b \rangle \approx 9$, $t_1 = \frac{6\pi}{\omega_0}$, $t_2 = \frac{41\pi}{\omega_0}$, $g_d = 0.151$, $g_c = 0.03$, $g_f = 0.01$ and $\omega_0 = 2.56$. (c,d) SHG case, starting from $|\Psi_0'\rangle$ with $\beta^2 = 9$, $g_c = 0.08$, $g_f = 0.01$ and $\omega_0 = 1.28$. The time-evolved plots are magnified for visual clarity, and in all cases $U = 1$ and $r_0 = 1.156$.

## 5 Molecular dissociation and optical response

Fig. 4 shows the fluorescence spectra for finite $M$, with all the elements previously discussed (photon pumping speed, atomic dynamics and cavity leakage) at interplay. The spectra in panels (a,b) and (c,d) respectively correspond to the atomic probabilities $N(t, r)$ of Fig. 3a and Fig. 3b. At resonance, the fluorescence spectrum strongly depends on the value of the atomic mass: For $M = 40$ the molecule dissociates (see Fig. 3a) and $\mathcal{P}(t, \omega)$ exhibits sharp features as well as a plateau, in stark difference to the Mollow-like structure of the rigid molecule limit. Conversely, for $M = 8 \times 10^4$, the molecule remains localized around the equilibrium position (inset in Fig. 3a), and at long times $\mathcal{P}(t, \omega)$ is peaked around the resonant value ($\Omega_R = 2.56$). Overall, the shape of $\mathcal{P}(t, \omega)$ with or without the bath dissipation show a mutual resemblance at long times. However, for bath dissipation the intensity of $\mathcal{P}(t, \omega)$ is considerably weaker.

A quite different picture emerges for SHG regime (Fig. 4c and d), where $\mathcal{P}(t, \omega)$ is considerably weaker in the case of an oscillator bath. Also, when the molecule dissociates (Fig. 4c), the SHG signal is absent irrespective of the presence or not of the baths. Conversely, for larger $M$, the SHG signal is present if the system evolves in contact with an oscillator bath, but with

smaller intensity. This suggests that the multi-photon cavity field is much more affected by dissipation under off-resonant conditions than at resonance.

In summary, in the dissociation regime both resonant Mollow and SHG signals are quenched. Also, for dissipation via an oscillator bath, for a broad range of atomic mass values fluorescence is always vastly reduced. Finally, even with no cavity leakage, the strength of the SHG response is determined by the cavity pumping rate.

# 6  Conclusion

Many decades of nonlinear optics research gave us a robust conceptual understanding of SHG, and actual uses in technology. Yet, some SHG regimes remain little explored, and how different physical mechanisms and interactions contribute to fluorescence is not always understood. In this work, we studied theoretically one of these (namely, the low photon) regimes, using a model molecule in an optical cavity, and via an exact time-dependent configuration interaction (TDCI) approach, where all quantum degrees of freedom (electrons, photons and relative atomic motion) are included on equal footing and supplemented by a semi-classical treatment of cavity dissipation/leakage.

Our study reveals a previously unknown, complex landscape for fluorescence, where the latter is reduced by electronic interactions and by cavity leakage, enhanced by fast cavity pumping, and quenched by molecular photodissociation. These competing trends likely occur in real molecules as well; it should thus be possible to detect them in experiments at low photon regimes. Our theoretical and computational framework can be applied and extended in different ways, e.g.more realistic molecules, or cavities with more than one molecule. Other possibilities are few ultracold bosons in cavities, to provide insight for SHG in the Gross-Pitaevskii limit [48], or fermions in the (interacting) Dicke's model, in conjunction with other techniques that exhibit better size-scaling behavior than TDCI, e.g. nonequilibrium Green's functions [49]. Some of these undertakings are under way.

# Acknowledgements

We acknowledge A. D'Andrea for discussions.

**Author contributions**  M.G. performed all calculations and interpretation of results under the supervision of E.V.B. and C.V. The project was conceived by E.V.B. and C.V. The overall supervision of the project was by C.V. Both M.G. and E.V.B. contributed to the writing of the code. All authors collaborated in writing the paper.

**Funding information**  M.G. and C.V. acknowledge support from the Swedish Research Council (grants number 2017-03945 and 2022-04486).

# A Further details and additional results

## A.1 Resonant frequency for the dimer molecule

To discuss the selection rules for light absorption, it suffices to consider a fixed molecule. The relevant part of the molecule Hamiltonian in this case is

$$H_e = -V_{eff}^{eq} \sum_\sigma \left( \hat{c}_{1\sigma}^\dagger \hat{c}_{2\sigma} + \hat{c}_{2\sigma}^\dagger \hat{c}_{1\sigma} \right) + U \sum_{i=1,2} \hat{n}_{i+} \hat{n}_{i-}, \tag{A.1}$$

where $V_{eff}^{eq} > 0$. The molecule-light interaction for the two photon modes is taken as $\hat{H}_{\text{int}}(t) = \hat{\mathcal{M}} \left[ g_c(b^\dagger + b) + g'(t)(b'^\dagger + b') \right]$, where $\hat{\mathcal{M}} = \sum_\sigma (c_{b\sigma}^\dagger c_{a\sigma} + c_{a\sigma}^\dagger c_{b\sigma})$. For two electrons of opposite spin, $H_e$ has three singlet eigenstates ($S = S_z = 0$) and one triplet eigenstate ($S = 1, S_z = 0$). The eigenvalues are 0 for $S = 1$ and $U, U/2 \mp \sqrt{4(V_{eff}^{eq})^2 + (U/2)^2}$ for $S = 0$. The ground state is the singlet with energy $U/2 - \sqrt{4(V_{eff}^{eq})^2 + (U/2)^2}$, and it is even under spatial parity. The eigenstates with odd symmetry under parity have energies 0 with $S = 0$ and $U$ with $S = 1$.

It can be easily shown that optical transitions between the two even ($E$) many-body states or between the two odd ($O$) many-body states are forbidden (e.g. $\langle E_1 | \hat{\mathcal{M}} | E_2 \rangle = 0$), and the only permitted transitions are between odd and even ones (i.e. with opposite parity). Furthermore, using the matrix expressions above for $\hat{\mathcal{M}}$ and $\hat{\mathbf{S}}^2$, one can show that $[\hat{\mathcal{M}}, \hat{\mathbf{S}}^2] = 0$. So the only transition allowed from the ground state is the even-odd one where the system goes $|g, S = 0\rangle \rightarrow |O, S = 0\rangle$ and where the energy difference is $\Omega_R = E_{O,S=0} - E_{g,S=0} = U/2 + \sqrt{4(V_{eff}^{eq})^2 + (U/2)^2}$, which defines the "many-body" resonance condition for the $\omega_0$ field in perturbation theory, similar to the two-level single-particle case. More in general, for the multi-photon case of interest here, the bare electronic many-body levels are renormalised by the photons, parity gets mixed up, and more transitions are possible and, most importantly, the parity of the full electron+photon systems must be considered. In the presence of nuclear dynamics, the values of the effective hopping parameter in the dimer changes in time and so it does $\Omega_R$.

## A.2 The interaction parameters

Before choosing the values for the parameters $g_c$, $g_f$ and $U$ used in the paper, we have performed calculations to observe their effect on the spectra. A sample of the ensuing results is reported in Fig. 5. Due to coupling between light and the molecule, the molecular levels will split and the splitting energy is $\propto g_c$ [37]. Hence the regime of the emitted photon frequency will be affected by the incident field coupling, as observed in Fig. 5. On increasing $g_c$, the fluorescent spectra get broadened, since this involves large range of frequencies for the emitted photon. On the other hand, Increasing the coupling $g_f$ increases the intensity of the fluorescent spectra. The electron interaction $U$ hinders electronic hopping between the two sites of the molecule. The emission of the fluorescent photon requires a transition among bonding and the anti-bonding molecular levels, and thus it involves electron hopping between the molecular sites. Accordingly, increasing the electron interaction decreases the intensity of the emitted photon, as it can be observed in Fig. 5.

## A.3 Pumping rate and resonant regime for a two-level system

In Fig. 6, we show $\mathcal{P}(\omega)$ for $\omega_0 = \Omega_R$ for two driving speeds as well as for photons initially in a coherent state. We observe similar trends as in the SHG regime discussed in Fig. 2, namely fast pumping leads to closer agreement with the coherent state spectrum. Since photons interact

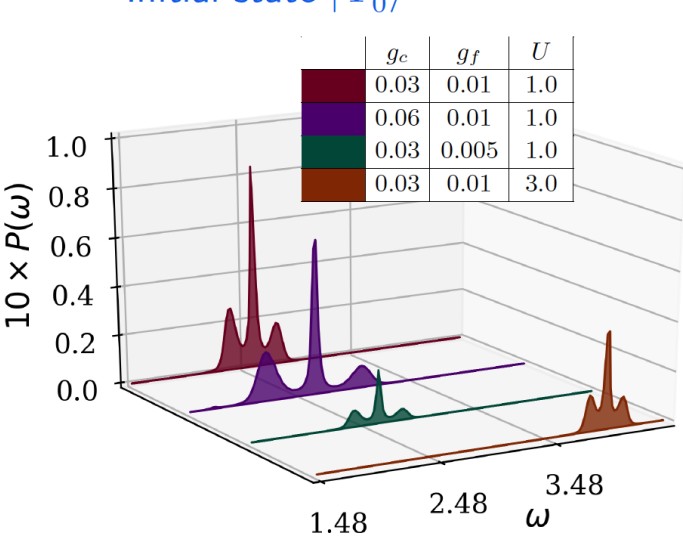

Figure 5: Fluorescent spectra for the rigid molecule starting with a coherent state with with $\beta = 3$, $r_0 = 1.156$, $\Gamma = 0.02$, $\Omega_R = 2.56$ and $\omega_0 = \Omega_R$. Only for the $U = 3.0$ case, it is $\Omega_R = \omega_0 = 3.95$ and $r_0 = 1.213$.

with the TLS during the drive, the coherent and fast-drive spectra become increasingly similar when the system-cavity interaction $g_c$ is decreased.

## A.4 Cavity leakage via a Caldeira-Leggett bath: Some details

To introduce leakage in the cavity, we use ideas borrowed from the physics associated with the Caldeira-Leggett model (CLM). As specified in Eq. (5) of the main text, we connect each the two modes $\omega_0$ and $\omega$ to a bath of $N_B$ classical oscillators. In the following, however, to provide details about the procedure, we consider for simplicity only one mode, say the $\omega_0$ incident mode. The case of the second (fluorescent) mode can be treated similarly.

The classical version of the CLM is defined as

$$H = \frac{\tilde{p}^2}{2\mu} + V(x) + \sum_{k=1}^{N_B} \left[ \frac{p_k^2}{2m_k} + \frac{1}{2} m_k \omega_k^2 \left( x_k - \frac{C_k}{m_k \omega_k^2} x \right)^2 \right], \tag{A.2}$$

where $\tilde{p}^2/2\mu + V(x)$ is the system (particle) Hamiltonian and the bath degrees of freedom are represented by the $2N_B$-tuple $\{x_k, p_k\}$. The oscillators have masses and frequencies $\{m_k, \omega_k\}$, and the coefficient $\{C_k\}$ determine the interaction between the particle and the bath. The form of the interaction term is chosen in this way to ensure translational invariance of the model in some specific situations [45].The solution of Eq. (A.2) can be written as

$$\mu \ddot{x}(t) + \frac{dV}{dx} + \mu \int_{t_0}^{t} \gamma(t - t') \dot{x}(t') dt' = -\mu \gamma(t - t_0) x(t_0) + F_L(t), \tag{A.3}$$

where $\gamma(t)$ determines the dissipative features of the bath (for example, for $\gamma(t) \to \gamma_0 \delta(t)$, we have a standard friction term), and $F_L(t)$ is a noise-like, oscillating force coming from the bath degrees of freedom. In the continuum-bath limit, $\gamma$ takes the form

$$\gamma(t) = \frac{2}{\pi} \int \frac{J(\omega)}{\mu \omega} \cos \omega t \, d\omega, \tag{A.4}$$

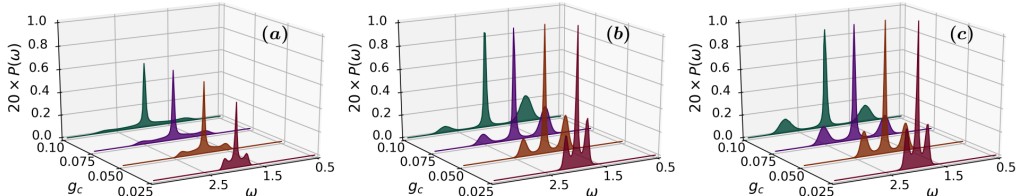

Figure 6: Long-time limit of the fluorescence spectra for a two-level system in the resonant regime and in a pumped cavity. Panel a): Pumping with $t_1 = 6\pi/\omega_0$ and $t_2 = 31\pi/\omega_0$. With reference to the values of $g_c$ in increasing order in figure, the corresponding used values of $g_d$ respectively are 0.212, 0.218, 0.224, and 0.232. Panel b): Pumping with $t_s = \pi/\omega_0$ and $g_d = 5.150$ for all $g_c$ cases. For both a) and b) panels, the pumping is applied until $\langle b^\dagger b \rangle \approx 16.0$ in the cavity and the initial state is $|\Psi_0''\rangle$. Reference results starting the time evolution from an initial coherent $|\Psi_0'\rangle$ state with $\langle b^\dagger b \rangle = 16.0$ but without pumping are also shown (c). Spectral intensities are in arbitrary units, and parameters common to all panels are $g_f = 0.01$, $\Gamma = 0.02$ and $\omega_0 = \Omega_R = 2.0$.

where $J(\omega) = \frac{\pi}{2} \sum_{k=1}^{N_B} \frac{C_k^2}{m_k \omega_k} \delta(\omega - \omega_k)$ is the spectral density of the bath (here, and in the rest of this section, $\omega$ is the variable Fourier-conjugated to $t$). Often, in practice, one takes $J(\omega) \propto \omega^\alpha$ in an interval range $[0, \omega_c]$, and zero otherwise. To describe the leaking from the cavity, we adopt a modified form of the CLM, where the photon modes are in the second quantisation picture, the masses $\mu = m_k = 1$, and we neglect the zero energy of the photon modes. Specialising to the mode $\omega_0$, this gives

$$\hat{H}_{1mode}(t) = \left( \omega_0 + \sum_{k=1}^{N_B} \frac{C_k^2}{\omega_k^2} \right) b^\dagger b + \sum_{k=1}^{N_B} \left( \frac{p_k^2}{2} + \frac{1}{2} \omega_k^2 x_k^2 \right) - \sum_{k=1}^{N_B} C_k x_k (b^\dagger + b)$$
$$+ \left[ \sum_{k=1}^{N_B} \frac{C_k^2}{\omega_k^2} \right] \frac{(b^\dagger)^2 + b^2}{2} + \hat{H}_{\text{mol}} + V_{\text{ext}}(t) + \hat{\mathcal{M}} g_c (b^\dagger + b), \tag{A.5}$$

where $\hat{\mathcal{M}}$ describes the electronic transitions. In this case, with $\tilde{C}_k = (2\omega_0)^{\frac{1}{2}} C_k$, we have $J(\omega) = \frac{\pi}{2} \sum_{k=1}^{N_B} \frac{\tilde{C}_k^2 \delta(\omega - \omega_k)}{m_k \omega_k}$. To choose the set $\{\tilde{C}_k\}$, we consider that, for a very large frequency $\omega_{Max}$, we get $\int_0^{\omega_{Max}} J(\omega) d\omega = \sum_{k=1}^{N_B} \frac{\tilde{C}_k^2}{\omega_k}$. By approximating the integral with a discrete sum with frequency step $\Delta$,

$$\sum_{k=1}^{N_B} \frac{\tilde{C}_k^2}{\omega_k} = \int_0^{\omega_{Max}} J(\omega) d\omega \approx \sum_{k=1}^{N_B} J(\omega_k) \Delta, \tag{A.6}$$

and thus $\frac{\tilde{C}_k^2}{\omega_k} \approx J(\omega_k) \Delta$. In turn, this amounts to say that [45, 46]

$$\frac{C_k^2}{\omega_k} (2\omega) \approx J(\omega_k) \Delta \Rightarrow C_k \approx \sqrt{J(\omega_k) \omega_k}. \tag{A.7}$$

The actual dynamics is performed according to the quantum-classical (Ehrenfest's) approximation, where the molecule+boson (m+b) system is quantum and the bath is classical. The equations of motion then are:

$$i \frac{d|\psi_{m+b}(t)\rangle}{dt} = \tilde{H}(\{x_k(t)\}) |\psi_{m+b}(t)\rangle, \tag{A.8}$$

$$\ddot{x}_k(t) = -\omega_k^2 x_k(t) + C_k(t) \langle b^\dagger + b \rangle_t, \tag{A.9}$$

$$\dot{x}_k = p_k, \tag{A.10}$$

where

$$\tilde{H}(\{x_k(t)\}, t) = \left(\omega_0 + \sum_{k=1}^{N_B} \frac{C_k^2}{\omega_k^2}\right) b^\dagger b - (b^\dagger + b) \sum_{k=1}^{N_B} C_k x_k(t) + \left[\sum_{k=1}^{N_B} \frac{C_k^2}{\omega_k^2}\right] \frac{(b^\dagger)^2 + b^2}{2} \quad \text{(A.11)}$$
$$+ \hat{H}_{\text{mol}} + V_{\text{ext}}(t) + \hat{\mathcal{M}} g_c (b^\dagger + b).$$

The Schrödinger equation (A.8) is solved as usual while for the bath fields we use the coordinate Verlet algorithm. In the actual calculations, the parameters were chosen such as $\omega_k = \Delta k$ and $C_k \propto k^a$. As it can be gathered from the foregoing discussion, the case of the fluorescent field in the presence of a bath can be treated similarly. The system of equations for the full system thus is

$$i \frac{d|\psi_{m+b}(t)\rangle}{dt} = \tilde{H}(\{x_k(t)\}, \{x_l'(t)\}, t)|\psi_{m+b}(t)\rangle, \quad \text{(A.12)}$$

$$\ddot{x}_k(t) = -\omega_k^2 x_k(t) + C_k(t)\langle b^\dagger + b\rangle_t, \qquad \dot{x}_k = p_k, \quad \text{(A.13)}$$

$$\ddot{x}'_l(t) = -\omega_l^2 x'_l(t) + C_l(t)\langle b'^\dagger + b'\rangle_t, \qquad \dot{x}'_l = p'_l, \quad \text{(A.14)}$$

where

$$\tilde{H}\left(\{x_k(t)\}, \{x_l'(t)\}, t\right) = \left(\omega_0 + \sum_{k=1}^{N_B} \frac{C_k^2}{\omega_k^2}\right) b^\dagger b - (b^\dagger + b) \sum_{k=1}^{N_B} C_k x_k(t) + \left[\sum_{k=1}^{N_B} \frac{C_k^2}{\omega_k^2}\right] \frac{(b^\dagger)^2 + b^2}{2}$$
$$+ \left(\omega + \sum_{l=1}^{N_B} \frac{C_l^2}{\omega_l^2}\right) b'^\dagger b' - (b'^\dagger + b') \sum_{l=1}^{N_B} C_l x'_l(t) + \left[\sum_{l=1}^{N_B} \frac{C_l^2}{\omega_l^2}\right] \frac{(b'^\dagger)^2 + b'^2}{2}$$
$$+ \hat{H}_{\text{mol}} + V_{\text{ext}}(t) + \hat{\mathcal{M}}\left[g_c(b^\dagger + b) + g'(t)(b'^\dagger + b')\right]. \quad \text{(A.15)}$$

## A.5  Frequency renormalization by the bath(s)

As seen in Appendix A.4, in the presence of baths the frequencies $\omega_0, \omega$ become renormalized, and an additional interaction contribution of the kind $(b^\dagger)^2 + b^2$ appears. The origin of these changes is easily understood looking at the classical CLM in Eq. (A.2): they are due to the contribution $\frac{1}{2}\sum_k \frac{C_k^2}{m_k \omega_k^2} x^2$, that in the quantum case behaves like $\approx (b^\dagger + b)^2$. As mentioned earlier, such term is present to ensure that the particle-bath interaction is translationally invariant, e.g. when $C_k = m_k \omega_k^2$ or when a coordinate transformation is performed. However, for the system considered here, this is a *non issue*: a (finite) cavity breaks translational invariance. However, since it is customary in the literature to consider the CLM as in Eq. (A.2), we wish to discuss here the role of this term for our molecule+cavity system. Similarly to Appendix A.4, we will carry out our analysis in terms of the $\omega_0$ mode only.

Let us to write again $H_{1mode}$ from Eq. (A.5), but more concisely:

$$\hat{H}_{1mode}(t) = (\omega_0 + A)b^\dagger b + A\frac{(b^\dagger)^2 + b^2}{2} \quad \text{(A.16)}$$

$$+ \hat{\mathcal{M}} g_c(b^\dagger + b) - \sum_{k=1}^{N_B} C_k x_k(b^\dagger + b) + \hat{H}_{\text{mol}} + \hat{H}_{\text{bath}} + V_{\text{ext}}(t), \quad \text{(A.17)}$$

where $A = \sum_k \frac{C_k^2}{\omega_k^2}$, and $H_{\text{bath}} = \sum_{k=1}^{N_B}\left(\frac{p_k^2}{2} + \frac{1}{2}\omega_k^2 x_k^2\right)$. Clearly, setting $A = 0$ in this expression is an approximation (it forces the removal of the quadratic terms). We also wish at this point to make explicit the form external potential:

$$V_{\text{ext}}(t) = g_d f(t) \sin(\omega'' t)(b^\dagger + b) \equiv g_d^{\omega''}(t)(b^\dagger + b). \quad \text{(A.18)}$$

This is the same as in the paper, but with the notable difference that the frequency $\omega''$ is left unspecified (in the paper, $\omega' = \omega_0$ always). We can now proceed to a Bogolubov transformation $\begin{pmatrix} b \\ b^\dagger \end{pmatrix} = \begin{pmatrix} u & v \\ v & u \end{pmatrix} \begin{pmatrix} d \\ d^\dagger \end{pmatrix}$ of the terms of the first line of Eq. A.17, and rewrite

$$(\omega_0 + A)b^\dagger b + (A/2)\big[(b^\dagger)^2 + b^2\big] \rightarrow \Omega d^\dagger d, \quad \text{where } u/v = [\sqrt{\omega_0/\Omega} +/- \sqrt{\Omega/\omega_0}]/2,$$

and $\Omega = (\omega_0^2 + 2\omega_0 A)^{\frac{1}{2}}$. Extending the transformation to the other terms of $\hat{H}_{1mode}$, we finally arrive at

$$\hat{H}_{1mode} = \Omega d^\dagger d + \sqrt{\frac{\omega_0}{\Omega}}\bigg[ g_c \hat{\mathcal{M}} - \sum_{k=1}^{N_B} C_k x_k + g_d^{\omega''}(t)\bigg](d^\dagger + d) + \hat{H}_{\text{mol}} + \hat{H}_{\text{bath}}, \qquad (A.19)$$

where we used Eq. (A.18) for $V_{\text{ext}}(t)$, and where some constant term have been dropped. As shown in Appendix A.4 in a related context, the manipulations done here apply straightforwardly to the fluorescent mode.

As a final point, and specifically considering the incident mode, we observe that if in Eq. (A.19) we set $\omega'' = \omega_0$, we then go back to the slightly-off-resonance case studied in the paper, but described in another, exact, representation (we have verified numerically that this is the case).

However, if we imagine that $V_{\text{ext}}(t)$ describes a laser with a tuneable frequency, we see that at $\omega'' = \Omega$, we are again in resonance with a cavity with an effective frequency renormalised by the bath(s), which should reflect as usual into an enhancement of the signal. Quite interestingly, both at- and away-from-resonance the problem can be described with a CLM bath without a quadratic term in $(d^\dagger + d)^2$, i.e. the translational invariance requirement does not play explicitly a role.

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
