# Peer review of "Photon pumping, photodissociation and dissipation at interplay for the fluorescence of a molecule in a cavity"

_SciPost Physics, doi:SciPost Phys. 15, 138 (2023)_

## Round 1 · Referee Report · Anonymous (Referee 2) · 2023-3-31

Strengths

1-The model accounts for different competing light-matter phenomena: leakage, photon confinement, nuclear motion and electronic correlations.
2-Exact time-dependent treatment of all degrees of freedom.

Weaknesses

1-The molecular model is quite limited.
2-The interatomic interactions seem unrealistic.
3-Relevance of the model unclear.

Report

In this work Gopalakrishna et al, develop a simple model for a one dimensional molecular dimer coupled to cavity photons and driven by a fluorescent field. Despite its simplicity on the molecular side, the model has a diverse character as it incorporates several different competing light-matter phenomena and mechanisms, like the confinement of photons via the cavity, photon pumping, photon leakage, nuclear motion and electron interactions. All degrees of freedom are treated quantum mechanically via an exact time-dependent configuration interaction approach.

The most important findings of this work are: (i) the second harmonic generation becomes larger with faster photon pumping. (ii) The electronic interactions suppress the fluorescence signal. (iii) The fluorescent and second harmonic generations signals depend significantly on the atomic mass and on the cavity leakage.

The manuscript it is definitely interesting and well-written. It incorporates many different aspects of light-matter interaction and several interesting phenomena take place as I summarized above. However, in my opinion the model molecule is clearly limited, it is one-dimensional which implies that rotational degrees of freedom are not taken into account, but most importantly only a single orbital per atom/site is included which essentially allows only a hopping to take place for the electrons between the two atoms/sites. Thus, the relevance of the model for realistic systems to me it is not obvious but rather unclear.

Further, the significance of the results is not evident to me. In my opinion this work does not meet the criteria for publication in Scipost Physics. I do not see how the results of this work can be considered as groundbreaking or opening a new pathway or a new research direction.

In my opinion, this is an interesting work which fits better to the publication criteria of Scipost Physics Core. I could recommend publication in Scipost Physics Core once the following points are adequately addressed.

Requested changes

1) The Hamiltonian, in the un-numbered equation, where the model is introduced it is written in a mixture between first and second quantized notation. This is rather unusual and it makes difficult to intuitively understand the model. It would be beneficial for the accessibility of the work a pure first or second quantized definition of the Hamiltonian to be given. I am particularly referring to the fact that the matter part is at the same time described with momentum and position operators in configuration space and fermionic annihilation and creation operators.

2) The un-numbered equation should be numbered as it is the defining equation of the model.

3) The interatomic potential used in the model is of the form 1/x^4. However, interatomic potentials typically are Lennard-Jones or Morse potentials. None of these potentials has such a form. The authors should clarify why they use the 1/x^4 potential, it seems rather unphysical. Also they should clarify to what extent the obtained results depend on the form of the chosen potential.

4) For the treatment of the bath the authors employ the well-known Caldeira-Leggett model. As they describe in Appendix A.4, the authors drop the quadratic ~(x)^2 term from the Caldeira-Leggett model, and they treat only the bilinear coupling between the molecule and the bath. However, dropping the quadratic term violates the translational invariance of the model, rendering their results dependent on the reference frame. To put differently performing a translation r -> r+a the model Hamiltonian does not remain invariant. This makes questionable the obtained results. The authors need to substantiate the validity of their approximation or they should include the quadratic term.

---

## Round 1 · Referee Report · Anonymous (Referee 1) · 2023-3-31

Report

In this interesting manuscript, Gopalakrishna et al. study second harmonic generation (SHG) in a model molecule in an optical cavity. The Authors employ an exact time-dependent configuration interaction method to include all quantum degrees of freedom (electrons, photons and relative atomic motion) on equal footing. Even if the description is for a model molecular system, the theoretical formulation at the exact level is nonetheless impressive. The reported findings shed light on several optical response scenarios: Electron-correlation effects on Mollow spectra, a competition between SHG and photo-dissociation, and photon-pumping effects on fluorescence.

While the methodological aspects and the reported findings certainly deserve to be published, the manuscript in its present form contains some issues (mostly straightforward), which would require a revision before I can recommend publication in SciPost Physics. I elaborate my observations below.

(1) The manuscript has been structured with an appendix for technical details to better streamline the core message. This is fine as long as important concepts are discussed so that the development is properly justified. For example, the total Hamiltonian contains external fields V_ext, for which the Authors write: ''will be specified later''. However, starting on line 96 the Authors already state time-scale arguments which would require knowing something about the external fields. The specific form of the driving is given only later as a footnote [45], which in my opinion would deserve a proper description in the main text since it relates to, e.g., the accumulated photon number (central concept).

(2) The photon Hamiltonian on line 88 relates omega to fluorescent photons. In Eq. (1) these are referred to as omega'. This is of course a choice (and a very minor issue) but it would be helpful to remain consistent. The primed symbols were generally linked to the fluorescent part. The spectrum P is also sometimes written P(w,t) and sometimes P(t,w).

(3) The fluorescent coupling is studied in two ways: phenomenologically with an exponential damping ~exp(-Gamma*t) and more systematically with a bath of harmonic oscillators. I feel this comparison is an important contribution (as many studies have been conducted using the phenomenological approach), and the agreement in Figure 3 shows that the value Gamma=0.02 is ''appropriate'' in this situation. Is there a way to estimate the strength of Gamma from the Caldeira-Legget bath? [See also point (7) below.]

(4) The Authors mention they have ''ensured numerical convergence with respect to these parameters'' referring to the number of cavity photons, fluorescence photons, and nuclear motion grid points. It would be interesting to know from a computational point of view what these numbers are roughly to appreciate the computational cost.

(5) Figure 1 discussion in the text refers to panels a-d but the figure contains only panels a-b. The associated results are distinguished with empty/filled curves. In the Figure 1 caption there could be some punctuation between 9 and t_1 so the photon number does not get mixed with the pumping time parameter. In relation to my first point, here it is difficult to appreciate the role of g_d.

(6) Figure 2 and 6 references seem to have been mixed. The text in section 3.1 refers to Figure 6 although it is clear Figure 2 is described. Regarding Figure 2, the Authors write about the excited state populations: ''very small for the pumped cases, but noticeably large for the coherent case''. I do not think the figure really supports this. It is true that in the coherent case n_1 is about twice as large as in the pumped case. To further assess the validity of this statement, in Figure 6 (Appendix A.3), it is not so easy to compare the peak heights.

(7) Is it not possible (or reasonably justifiable) to try to obtain the long-time limit of the fluorescence spectra for the Caldeira-Legget bath? I understand the motivation to compare the peak positions in Figure 4 with the long-time limit of the exponential dissipation. The peak-height comparison however seems more difficult: The Authors write ''for bath dissipation the intensity of P(w',t) is considerably weaker''. In this regard, is there some insight why the Mollow triplet is not as prominent (or not there at all) for the harmonic bath?

(8) The Authors conclude by a discussion on scaling their development up to more realistic systems. It might be useful for the reader to get an idea what this would amount to: E.g., what are the Gross-Pitaevskii limit or nonequilibrium Green's functions (electronic, photonic etc). Since this is probably not the point of this manuscript, I think references to literature would suffice.

---

## Round 1 · Referee Report · Anonymous (Referee 3) · 2023-4-4

Report

In this paper the Authors study theoretically the time-resolved fluorescence spectrum of a model molecule placed inside an optical cavity. Many effects are simultaneously considered, including electron-electron and electron-nuclear interactions, the nuclear motion, the quantum nature of the light, and the coupling of the system with a Caldeira-Leggett bath. The presented results are obtained within a virtually exact time-dependent numerical methods based on configuration interaction. The Authors are able to assess, among the several and complex competing effects, what are the physical processes that tend to enhance or quench the fluorescence emission. In particular they find that fluorescence is enhanced by a fast pumping rate while is suppressed by cavity leakage, electronic correlations and molecular dissociation.
The manuscript is well written and the results are presented in a clear and comprehensive way, with the help of explanatory figures, and by including additional details in the appendices.

Despite the model is very simple, exact results are highly valuable as they serve as a reference point to develop and benchmark approximations for the study of more realistic systems where the exact solution is out of reach. For this reason the presented work can potentially guide future works on the topic, and I therefore recommend it for publication after the Author address the minor points listed below.

1) The model contains several parameters. While I understand that it is not possible to present a systematic study across the whole phase space, the Authors should explain why the chosen values of the model parameters are relevant to capture the qualitative behavior of a realistic molecule in an optical cavity.

2) The Caldeira-Leggett bath responsible for the cavity leakage is treated within the Ehrenfest approximation. The Authors should discuss the validity of this approximation as it has been shown to violate the principle of detailed balance. This is could be relevant when considering the energy transfer from the cavity modes to the bath.

3) As one of the main aspects of the paper is the study of fluorescence and SHG spectra on the pumping rate, the Authors should show and discuss explicitly the pumping protocol that they have considered. They only mention the value of the pumping speed.

---

## Round 2 · Referee Report · Anonymous (Referee 3) · 2023-7-21

Report

The Authors have fully addressed all points raised by the three Referees.
I recommend publication of the manuscript in its present form

---

## Round 2 · Referee Report · Anonymous (Referee 1) · 2023-8-2

Report

My initial view on this manuscript was already quite positive. In their re-submission, the Authors have thoroughly addressed all my remarks, and I am thus happy to recommend publication of the manuscript in SciPost Physics. In particular, I appreciate the Authors' detailed answer and added discussion on the distinction between different dissipation/damping mechanisms.

---

## Round 2 · Referee Report · Anonymous (Referee 2) · 2023-8-3

Report

The authors in the revised manuscript and the corresponding reply letter have made clear the novelty and significance of their work. In addition, they have addressed the points I raised in the first review round, they have substantially modified and improved their manuscript. Thus, I recommend the publication of the paper in Scipost Physics.

---

## Round 2 · Author Response

—————— General premise to all Referees and to our answers to the reports —————

We would like to start our reply to the Referees by thanking them for their positive evaluation of the manuscript, and for their insightful and helpful remarks, criticisms and suggestions.

In working at the revision of the paper, and while addressing the Referees' suggestions for changes, we found that even in the presence
of oscillator baths, our numerical calculations still contained by mistake an exponential damping of the fluorescence field. Without such exponential phenomenological damping, the results of our original Fig.4 would be different. On a related matter, we were also asked by one of the Referees to introduce a so-called "counterterm" in the Caldeira-Leggett model (CLM), which ensures the translational invariance of the model. While not strictly necessary for a (finite) cavity, which has no translational invariance, we have nonetheless included this correction in the formalism and implemented it numerically. The presence of this term also changes, in an independent way, the results of the original Fig.4, essentially because the incident field is renormalized by the bath and goes off-resonance.

These matters are discussed now in two new Appendixes, where we also address in detail the CLM counterterm, and show that in principle its effect can be re-adsorbed by redefining the parameters. That is, by some parameter rescaling, and keeping the exponential damping, the result of the original Fig. 4 could be reproduced even in the presence of the counterterm. However, in the paper, we decided to not do such rescaling, and let the incident field to be out of resonance. However, this off-resonance effect is not large, and with a slight modification of the bath strength, but still keeping the exponential damping for the fluorescent field, the result of the new Fig.4 in the revised paper are quite close to the previous ones. Furthermore, the results of Fig. 1,2 were never affected by this, since there are no baths in that case, and the changes introduced in Fig. 3 are practically hardly discernible.

The last general point we wish to consider is that, in the original version of the manuscript, exponential damping and baths looked as if they were alternative treatment of generic dissipation. However, after giving some careful consideration to this point, we have come to realize that these two formally different sources of dissipation are in fact distinct but not mutually exclusive. As stated now in the paper, the phenomenological damping acts on the coupling between the matter and fluorescent photons, to account in an effective way for the fact that the spontaneous emission into a photon continuum is described via a single effective mode. On the other hand, with the bath of harmonic oscillators, we describe a dissipation channel for the photon modes, i.e. for the finite cavity quality. Since the photon-photon coupling utilised with the harmonic bath can be seen as an effect of all photon modes interacting via the molecular system, the two effect are clearly related, and yet rather distinct. So, it is appropriate and meaningful to keep these two dissipation mechanisms together when cavity leakage is present.

In addressing one by one the specific points in the Referees’ reports, we will refer to the n-the Referee’s point by “Rn” and to our reply to that point by “An”.

—————— Reply to Report 1 ——————

………….
General remarks from the Referee.-
In this interesting manuscript, Gopalakrishna et al. study second harmonic generation (SHG) in a model molecule in an optical cavity. The authors employ an exact time-dependent configuration interaction method to include all quantum degrees of freedom (electrons, photons and relative atomic motion) on equal footing. Even if the description is for a model molecular system, the theoretical formulation at the exact level is nonetheless impressive. The reported findings shed light on several optical response scenarios: Electron-correlation effects on Mollow spectra, a competition between SHG and photo-dissociation, and photon-pumping effects on fluorescence.

While the methodological aspects and the reported findings certainly deserve to be published, the manuscript in its present form contains some issues (mostly straightforward), which would require a revision before I can recommend publication in SciPost Physics. I elaborate my observations below.

** Authors **
We wish to thank the Referee for their appreciation of our work and for making important suggestions to improve the manuscript. We now proceed to address the specific remarks from the Referee.

………….
R1.-
The manuscript has been structured with an appendix for technical details to better streamline the core message. This is fine as long as important concepts are discussed so that the development is properly justified. For example, the total Hamiltonian contains external fields $V_{\rm ext}$, for which the authors write: ``will be specified later''. However, starting on line 96 the authors already state time-scale arguments which would require knowing something about the external fields. The specific form of the driving is given only later as a footnote [45], which in my opinion would deserve a proper description in the main text since it relates to, e.g., the accumulated photon number (central concept).

R2.- The photon Hamiltonian on line 88 relates omega to fluorescent photons. In Eq. (1) these are referred to as $\omega'$. This is of course a choice (and a very minor issue) but it would be helpful to remain consistent. The primed symbols were generally linked to the fluorescent part. The spectrum $P$ is also sometimes written $P(\omega,t)$ and sometimes $P(t,\omega)$.

** A1,A2 **
We thank the Referee for pointing out these inconsistencies, and to suggest how improve the readability and logical structure of the manuscript. We have incorporated the inherent changes, and the manuscript should now be straightforward to follow.

………….
R3.-
The fluorescent coupling is studied in two ways: phenomenologically with an exponential damping $\sim \exp(-\Gamma t)$ and more systematically with a bath of harmonic oscillators. I feel this comparison is an important contribution (as many studies have been conducted using the phenomenological approach), and the agreement in Figure 3 shows that the value $\Gamma = 0.02$ is ``appropriate'' in this situation. Is there a way to estimate the strength of Gamma from the Caldeira-Legget bath? [See also point (7) below.]

** A3 **
We thank the Referee appreciating our modeling of cavity dissipation via classical harmonic oscillators. In fact, the roles of the phenomenological damping $\Gamma$ and the harmonic oscillators are not exactly equivalent. The former acts on the coupling between the matter and fluorescent photons, and accounts in an effective way for the fact that the spontaneous emission into a photon continuum is described via a single effective mode. In contrast, the harmonic oscillators act as a dissipation channel for the photon modes an thereby account in an effective way for the finite cavity quality. However, the two mechanisms are clearly related, since the photon-photon coupling utilised with the harmonic bath can be seen as an effect of all photon modes interacting via the molecular system.

The difference between the mechanisms can be seen by considering the long-time behavior of the model. With the phenomenological damping $\Gamma$ the number of photons in the fluorescent field will tend to a finite constant value $n_{\rm fl}^\infty$ as $t \to \infty$, and $\Gamma$ may be understood (approximately) as the inverse of the time where $n_{\rm fl}(t) \approx n_{\rm fl}^\infty/2$.

In contrast, for a coupling to a harmonic bath, the number of photons in the fluorescent field never becomes a strictly conserved quantity. Instead, $n_{\rm fl}^\infty$ will attain a steady-state value determined by the equation $\Gamma_{\rm in} = \Gamma_{\rm out}$, where $\Gamma_{\rm in}$ is the effective rate at which fluorescent photons are created and $\Gamma_{\rm out}$ is the effective rate at which fluorescent photons decay.

To get a rough estimate of the time it takes to reach the steady state, we assume that the number of photons in the cavity field is large compared to the number of photons generate in the fluorescent field. The creation rate is then expected to be constant and of the order $\Gamma_{\rm in} \sim g_{\rm fl}^2/\Omega_{\rm Rabi}$, where $\Omega_{\rm Rabi}$ is the Rabi frequency of the molecular system. Similarly the decay rate is expected to be of the order $\Gamma_{\rm out} \sim (\sum_k C_k^2/\omega_k) n_{\rm fl} = \Gamma_{\rm out}^0 n_{\rm fl}$, as set by the coupling to the classical oscillators. Within this simple approximation the fluorescent field will approach the $n_{\rm fl} = \Gamma_{\rm in}/\Gamma_{\rm out}^0$ as $n_{\rm fl} = (\Gamma_{\rm in}/\Gamma_{\rm out}^0) (1 - e^{-\Gamma_{\rm out}^0 t})$, such that the effective phenomenological damping rate can be approximated by $\Gamma_{\rm out}^0$.

Since the estimate just provided will be affected by many additional details of the model, we have chose not to include this discussion in the main text.

………….
R4.-
The Authors mention they have ``ensured numerical convergence with respect to these parameters'' referring to the number of cavity photons, fluorescence photons, and nuclear motion grid points. It would be interesting to know from a computational point of view what these numbers are roughly to appreciate the computational cost.

** A4 **
In the model, the parameters determining the size of the quantum problem (the dynamics of the $N_B$ classical oscillators is inexpensive) are the number of: photons of the incident field ($N_c$), photons of the fluorescent field ($N_f$), nuclear coordinate gridpoints ($N_R$).The size of the fermion part is fixed (=4 for the Hubbard dimer in the $S_z=0$ subspace). For all calculations,
$N_c \lesssim 65$, $N_f=5$ (since the fluorescence signal is very weak), $500 \lesssim N_R \lesssim 1000$.
For all of these parameters, we have done calculations with values, say, $N$ and $N+n$, with $n\approx 5\%$ of $N$,
with the results unchanged. So, the overall configuration maximum size was around $N_{tot}\lesssim 1.3 \times 10^6$.

………….
R5.-
Figure 1 discussion in the text refers to panels a-d but the figure contains only panels a-b. The associated results are distinguished with empty/filled curves. In the Figure 1 caption there could be some punctuation between 9 and $t_1$ so the photon number does not get mixed with the pumping time parameter. In relation to my first point, here it is difficult to appreciate the role of $g_d$.

** A5 **
We have amended the figure and the inherent discussion as suggested by the Referee. We thank the Referee
for noting these inconsistencies. Concerning the role of $g_d$: the fluorescence curves (filled curves) are obtained starting from the exact molecule+cavity ground state and then pumping photons (i.e. $g_d\neq 0$) in the cavity. These curves have much less intensity than the empty curves (note the scaling factors). The latter are obtained starting with a coherent photon state in the cavity + the molecule in its ground state, but applying no external field (i.e. $g_d =0$). This is how $g_d$ affect the results and this is what motivated the subsequent section in the paper. We have readjusted locally the text in the paper, to make this point clearer.

………….
R6.-
Figure 2 and 6 references seem to have been mixed. The text in section 3.1 refers to Figure 6 although it is clear Figure 2 is described. Regarding Figure 2, the Authors write about the excited state populations: ``very small for the pumped cases, but appreciably large for the coherent case''. I do not think the figure really supports this. It is true that in the coherent case $n_1$ is about twice as large as in the pumped case. To further assess the validity of this statement, in Figure 6 (Appendix A.3), it is not so easy to compare the peak heights.

** A 6 **
We thank the Referee for pointing out the incorrect reference to Fig,6, this now is fixed. Looking at the figure, the pumped populations are larger in a clear and distinct way, so we have changed the sentence to "...but noticeably larger for the coherent case...". About the second remark of the Referee: in Fig. 6a, the pumping is "slow", in Fig.6b is about 25 times faster, and in 6c is infinitely faster (i,e. we have an initial coherent state resulting from an infinitely fast ramping). As result, since the intensity scale is the same in all figures 6a,b,c the curves (and the intensities), it is apparent that in Fig. 6b and 6c the spectra are mutually very close, but they quite differ from those in Fig.6a.

………….
R7.-
Is it not possible (or reasonably justifiable) to try to obtain the long-time limit of the fluorescence spectra for the Caldeira-Legget bath? I understand the motivation to compare the peak positions in Figure 4 with the long-time limit of the exponential dissipation. The peak-height comparison however seems more difficult: The Authors write ''for bath dissipation the intensity of P(w',t) is considerably weaker''. In this regard, is there some insight why the Mollow triplet is not as prominent (or not there at all) for the harmonic bath?

** A7 **
The question posed by the Referee is very interesting. Unfortunately, the discussion of this issue is a bit conditioned by our mistake in our original calculations of Fig,4, i.e. that the exponential damping was present also when the bath was included. In view of this, we are not fully sure on how to address the original point from the Referee, about the original Figure. What we can see in our new calculations (which are now slightly off resonance, and with the exponential damping included) is that the Mollow triplet remains in the presence of leakage, and that, most likely, its intensity reduction has to do with the leakage that incident photons experience during the pumping. We have also noted (not shown in the paper) that in absence of exponential damping the systems seems to (very slowly) proceed towards a reduced signal, but with small superimposed irregular oscillations (these are washed out in the presence of exponential damping). However, the parameter space is so vast that we cannot exclude a priori that with different parameters the Mollow sideband(s) could be strongly depressed or altogether removed. Thus, we fully agree with the Referee that it could be interesting to look at the system's steady state in the presence of the bath. This should be done in the continuum-bath limit, generalising the approach in (Phys. Lett. A 180) 430 to a steady-state treatment that includes the classical oscillators. This development is beyond the scope of the present paper, and planned for future investigations.
Finally, we wish to thank the referee for this question, and we regret to not be able to give a more complete answer at this time.

………….
R8.-
The Authors conclude by a discussion on scaling their development up to more realistic systems. It might be useful for the reader to get an idea what this would amount to: E.g., what are the Gross-Pitaevskii limit or nonequilibrium Green's functions (electronic, photonic etc). Since this is probably not the point of this manuscript, I think references to literature would suffice.

** A8 **
By Gross-Pitaevskii (GP) limit, we meant the situation where a boson condensate in a cavity is described via the GP equation, which is an approximate, non linear, Hartree-like equation for the bosonic condensate. We have now put a reference to the GP equation, and for the nonequilibrium Green's functions (NEGF), we have put a reference to literature dealing with electron-boson systems.

%%%%%%%%%%%%%%%%%%%%%%%%%%%%%%%%%%%%
%%%%%%%%%%%%%%%%%%%%%%%%%%%%%%%%%%%%
%%%%%%%%%%%%%%%%%%%%%%%%%%%%%%%%%%%%
%%%%%%%%%%%%%%%%%%%%%%%%%%%%%%%%%%%%
%%%%%%%%%%%%%%%%%%%%%%%%%%%%%%%%%%%%
%%%%%%%%%%%%%%%%%%%%%%%%%%%%%%%%%%%%
%%%%%%%%%%%%%%%%%%%%%%%%%%%%%%%%%%%%
%%%%%%%%%%%%%%%%%%%%%%%%%%%%%%%%%%%%

—————— Reply to Report 2 ——————

………….
General remarks from the Referee.-
In this work Gopalakrishna et al, develop a simple model for a one dimensional molecular dimer coupled to cavity photons and driven by a fluorescent field. Despite its simplicity on the molecular side, the model has a diverse character as it incorporates several different competing light-matter phenomena and mechanisms, like the confinement of photons via the cavity, photon pumping, photon leakage, nuclear motion and electron interactions. All degrees of freedom are treated quantum mechanically via an exact time-dependent configuration interaction approach.

The most important findings of this work are: (i) the second harmonic generation becomes larger with faster photon pumping. (ii) The electronic interactions suppress the fluorescence signal. (iii) The fluorescent and second harmonic generations signals depend significantly on the atomic mass and on the cavity leakage.

The manuscript it is definitely interesting and well-written. It incorporates many different aspects of light-matter interaction and several interesting phenomena take place as I summarised above. However, in my opinion the model molecule is clearly limited, it is one-dimensional which implies that rotational degrees of freedom are not taken into account, but most importantly only a single orbital per atom/site is included which essentially allows only a hopping to take place for the electrons between the two atoms/sites. Thus, the relevance of the model for realistic systems to me it is not obvious but rather unclear.

Further, the significance of the results is not evident to me. In my opinion this work does not meet the criteria for publication in Scipost Physics. I do not see how the results of this work can be considered as groundbreaking or opening a new pathway or a new research direction.

In my opinion, this is an interesting work which fits better to the publication criteria of Scipost Physics Core. I could recommend publication in Scipost Physics Core once the following points are adequately addressed.

** Authors **
We thank the Referee for their appreciation of our work. However, we do not agree with the Referee's assessment of the novelty and impact of our work. The concrete criticisms presented by the Referee in this respect are that in our model we miss rotational degrees freedom, multiple orbitals, and higher dimensionality. This is true. But we have electronic interactions and dynamics, electron-nuclei and nuclei-nuclei interactions, cavity leakage, fluorescence, photodissociation, all in one computational box. To our knowledge, this is done here for the first time (hence the novelty), and we would like to hear from the Referee if we are mistaken. In short, ours is a minimum complication (but still highly complex) model where many different effect are treated simultaneously on equal footing and exactly in SHG, beyond supra-linear response schemes, and directly in the time domain. Certainly, multiple orbitals/atom would provide a richer picture, more available transitions, and the rotational degrees of freedom would give overtones in the SHG, and so even richer physics would emerge. This is in fact planned for future work. Other possibility are to characterize entanglement in the cavity, optimal control of the laser pulse, flying-by molecules in the cavity, to mention a few. This can all be as avenues for future extensions. So, clearly, here we do not want to simulate the experiment (not yet, at least). Rather, the important path our work can open (and thus its main impact) comes from suggesting (in fact, demonstrating) that, at the fundamental level, there is still so much to discover in harmonic generation (not only SHG; considering higher harmonics in our approach is only a matter of computational ''muscle") when we are in the low photon limit, where many photon effects are taken into account at the quantum level. And if, not totally unlikely, experiments will be stimulated by the implications/results of our work, that is exactly where the extensions suggested by the Referee will be very needed and definitely introduced. We hope that with these considerations we were able to clarify the novelty, aims and potential impact of our work.

………….
R1.-
The Hamiltonian, in the un-numbered equation, where the model is introduced it is written in a mixture between first and second quantized notation. This is rather unusual and it makes difficult to intuitively understand the model. It would be beneficial for the accessibility of the work a pure first or second quantized definition of the Hamiltonian to be given. I am particularly referring to the fact that the matter part is at the same time described with momentum and position operators in configuration space and fermionic annihilation and creation operators.

** A1 **
As noted by the Referee we employ a Hamiltonian written in a mixed first and second quantized representation. However, while this possibly is not the most usual representation, there are no principle problems with such an approach. Indeed, as will be discussed further below, this choice is highly favourable in the present case.

The basic reason: it is not very efficient to describe the dynamics of a strongly anharmonic nuclear potential using second quantization, since the typical vibrational picture relies on an expansion of the nuclear potential around some equilibrium. In particular, to capture a process like molecular dissociation would require operators in the Hamiltonian of infinitely high powers in the vibrational creation and annihilation operators. In contrast, such a process is very straightforward to simulate numerically on a grid. Likewise, the photon fields cannot be described using first quantization since the Hamiltonian involves photon creation and annihilation processes.

Therefore, the only part of the Hamiltonian where there is a choice in representation is for the electrons. Here, a first quantized description would require storing the two-particle electronic wavefunction on a two-dimensional grid, which very quickly becomes numerically inefficient. More specifically, the electronic subspace would scale as $N^2$ where $N$ is the number of grid points for a single electron. Even for a sparse grid with $N \approx 100$ this would require an electronic basis with $10000$ states. In contrast, working with an orbital representation as in our manuscript allows us to include the relevant orbitals using an electronic basis of four states (for our simple model molecule). For this reason, the mixed representation employed in our work is vastly superior to a pure first or second quantized approach.

………….
R2.-
The un-numbered equation should be numbered as it is the defining equation of the model.

** A2 **
We agree with the Referee that this equation should be numbered, and have changed this in the revised manuscript.

………….
R3.-
The interatomic potential used in the model is of the form $1/x^4$. However, interatomic potentials typically are Lennard-Jones or Morse potentials. None of these potentials has such a form. The authors should clarify why they use the $1/x^4$ potential, it seems rather unphysical. Also they should clarify to what extent the obtained results depend on the form of the chosen potential.

** A3 **
We thank the Referee for this comment, since it allows us to clarify some aspects of the interatomic potential employed. We would first like to note that most molecular potentials (including the Morse and Lennard-Jones ones) only aim at describing qualitative aspects of molecular binding, and often lack a clear physical justification. This is true for the Lennard-Jones potential where the $1/x^{12}$ is highly {\it ad hoc}, and is only used because of its numerical convenience (since it is the square of the $1/x^6$ term). In contrast, the $1/x^6$ has a physical justification as it has the behavior expected by dipolar van der Waals interactions. The same is true of the Morse potential which has an unphysical exponential divergence as $r \to 0$.

It is better then to ask what physical behavior we would expect of a molecular potential in a given physical context. In the present case, we would expect a polynomial repulsive term of the form $1/r^n$ arising from Coulombic nuclear-nuclear repulsion at short distances, and an exponentially decaying attractive term of the form $e^{-\lambda r}$ arising from the overlaps of the tails of the electronic orbitals localized on each of the ions. These considerations motivate the general form of our molecular potential, which is obtained by adding the repulsive and attractive parts. Regarding the choice of $n = 4$ in the inter-ionic repulsion, this is motivated from trade-off between strong repulsion and numerical convenience, since higher powers have stronger divergences as $r \to 0$.

Finally, we do not expect any of the qualitative conclusions obtained in our work to be sensitive to the precise form of the molecular potential. This is also our goal, since we do not target a specific molecular system. However, there will of course be quantitative differences with different potentials, and to address a given molecule a careful analysis of the molecular landscape should be performed.

………….
R4.-
For the treatment of the bath the authors employ the well-known Caldeira-Leggett model. As they describe in Appendix A.4, the authors drop the quadratic $\sim x^2$ term from the Caldeira-Leggett model, and they treat only the bilinear coupling between the molecule and the bath. However, dropping the quadratic term violates the translational invariance of the model, rendering their results dependent on the reference frame. To put differently performing a translation $r \to r+a$ the model Hamiltonian does not remain invariant. This makes questionable the obtained results. The authors need to substantiate the validity of their approximation or they should include the quadratic term.

** A4 **
We are extremely grateful to the Referee for raising the issue since, among others, it made us to go back to the code to implement the "counterterm" $\sim x^2$ and discover the mistake that the exponential damping was always present (see the general premise to our responses to all the Referees). Of course, we were already aware of the nature and purpose of the $\sim x^2$ term and that we were neglecting it (we made a statement in this regard in the original version of the manuscript). Our original thinking was that a cavity, if finite, breaks translational invariance, and so we were not concerned with translational symmetry in real space. This is certainly the case. However, motivated by the Referee, we looked much more carefully, and the result is some readjustment of our perspective on the role of the bath, and an extensive rewriting of one Appendix together and the addition of a new second one. In short, with the counterterm included, the bath renormalises the frequencies, and the cavity goes slightly (the bath interaction is not strong) off resonance with the molecular transition. This is the perspective adopted in the revised manuscript. However, we have also mathematically recasted the model in a way such that the counterterm formally exactly disappears, but all system-bath coupling parameters become slightly renormalized, in particular the frequency of the cavity modes. So, with a tuneable laser that can adjust the driving frequency to new cavity one, we are back to resonance but with a bath with renormalized parameters and without the counterterm. Thus, our summary response to the Referee is that i) neglecting the $\sim x^2$ term is not an approximation if one is in the right picture and has a tuneable laser, and some parameters are renormalized; and ii) we have included the counterterm in our revised calculations.

%%%%%%%%%%%%%%%%%%%%%%%%%%%%%%%%%%%%
%%%%%%%%%%%%%%%%%%%%%%%%%%%%%%%%%%%%
%%%%%%%%%%%%%%%%%%%%%%%%%%%%%%%%%%%%
%%%%%%%%%%%%%%%%%%%%%%%%%%%%%%%%%%%%
%%%%%%%%%%%%%%%%%%%%%%%%%%%%%%%%%%%%
%%%%%%%%%%%%%%%%%%%%%%%%%%%%%%%%%%%%
%%%%%%%%%%%%%%%%%%%%%%%%%%%%%%%%%%%%
%%%%%%%%%%%%%%%%%%%%%%%%%%%%%%%%%%%%

—————— Reply to Report 3 ——————

………….
General remarks from the Referee.-
In this paper the authors study theoretically the time-resolved fluorescence spectrum of a model molecule placed inside an optical cavity. Many effects are simultaneously considered, including electron-electron and electron-nuclear interactions, the nuclear motion, the quantum nature of the light, and the coupling of the system with a Caldeira-Leggett bath. The presented results are obtained within a virtually exact time-dependent numerical method based on configuration interaction. The authors are able to assess, among the several and complex competing effects, what are the physical processes that tend to enhance or quench the fluorescence emission. In particular they find that fluorescence is enhanced by a fast pumping rate while is suppressed by cavity leakage, electronic correlations and molecular dissociation. The manuscript is well written and the results are presented in a clear and comprehensive way, with the help of explanatory figures, and by including additional details in the appendices.

Despite the model is very simple, exact results are highly valuable as they serve as a reference point to develop and benchmark approximations for the study of more realistic systems where the exact solution is out of reach. For this reason the presented work can potentially guide future works on the topic, and I therefore recommend it for publication after the authors address the minor points listed below.

** Authors **
We wish to thank the Referee for their positive evaluation of our work, for making important suggestions to improve the manuscript, and for recommending publication. We now proceed to address the specific remarks from the Referee.

………….
R1.-
The model contains several parameters. While I understand that it is not possible to present a systematic study across the whole phase space, the authors should explain why the chosen values of the model parameters are relevant to capture the qualitative behavior of a realistic molecule in an optical cavity.

** A1 **
The parameters of our model can be divided into two categories, the molecular parameters $\{ V, U, M, \lambda, C \}$ and the photonic parameters $\{ \omega, \omega', g, g', \Gamma, \beta \}$. We begin with a discussion of the former.

For algebraic simplicity, let us momentarily work in effective units where the unit of energy is given by the effective electronic hopping at equilibrium $V_{\rm eff}= \langle |V| e^{\lambda r_0}\rangle$, and the unit of length by the molecular equilibrium interatomic distance $r_0 = 1$. The remaining molecular parameters can then be expressed as $U=V_{\rm eff}$, $\lambda = 0.6/r_0$, $C = 0.6 V_{\rm eff} r_0^4$ and $M/\hbar^2 = 40 V_{\rm eff}^{-1} r_0^{-2}$. We note that, according to this choice, the variables $\lambda$ and $C$ define the potential landscape of the molecular coordinate, and are fixed by requiring a plausible binding
energy $E_b$ and a reasonable shape of the lowest potential energy surface, plus an equilibrium hopping amplitude $V_{\rm eff} = \langle |V| e^{\lambda r_0}\rangle \approx 1$.

Translating these effective parameters into physical units, and setting typical molecular values $V_{\rm eff}= 1$ eV and $r_0 = 1$ {\AA}, we have $E_b = 1.2$ eV and $M = 305 m_e$ or $M = 334 m_u$, thus corresponding to the extreme ends of masses for a diatomic molecule. Alternatively, we can imagine an effective molecule made up of bound Wannier-Mott excitons, with typical values of $V_{\rm eff} = 10$ meV and $r_0 = 100$ {\AA} that would give a binding energy $E_b = 12$ meV and effective masses $M = 3 m_e$ or $M = 3 m_u$. Thus we see that typical values of the molecular parameters $V$ and $r_0$ give a reasonable parametrization of a diatomic molecule.

Coming to the actual values used in the paper, we have that the equilibrium $V_{\rm eff}= 1$. This is obtained by choosing $\lambda r_0=\ln 2$, $V=2V_{\rm eff}$, $\lambda=0.6$/\AA, to obtain $r_0 \simeq 1.156$\AA, with $E_b = 1.225 V_{\rm eff}$). Thus,
all previous estimate above, which were made with $r_0=1$ {\AA}, hold with few percents.

For the photonic parameters, we note that both $\omega$ and $\omega'$ for a molecule need to be in the eV range. Such frequencies are achievable in various microcavities [see e.g. Nature Photonics 9, 30–34 (2015); Nature Photonics 11, 491 (2017); Nature Nanotechnology 17, 1060 (2022)]. Similarly, for an excitonic realization the frequencies need to be in the THz (or few meV) regime, which is readily realizable e.g. via so-called split-ring resonators [ACS Photonics 5, 2, 278 (2018); ACS Photonics 8, 9, 2692 (2021)]. Finally, the light-matter couplings considered are on the order of $g = 0.01 - 0.1$, and are large for an isolated system but not unreasonable. In particular, considering a molecular gas at low density (such that intermolecular interactions are negligible) but coherently interacting with the photon fields, a collective coupling scaling as $\sim \sqrt{n}$ with the number of molecules $n$ reaching this order could be expected [J. Chem. Phys. 156, 230901 (2022)].

In conclusion, the parameters of the model give substantial freedom to consider different physical realizations, and provide a reasonable qualitative description of the discussed physical systems.

………….
R2.-
The Caldeira-Leggett bath responsible for the cavity leakage is treated within the Ehrenfest approximation. The authors should discuss the validity of this approximation as it has been shown to violate the principle of detailed balance. This is could be relevant when considering the energy transfer from the cavity modes to the bath.

** A2 **
We thank the Referee for raising this point. Indeed, there could in principle be a problem with detailed balance, a thing which is well known for e.g. the case of electron-nuclei systems (J. Phys.: Condens. Matter 16 3609), electron-spin interactions (Phys. Rev. Lett. 119 (227203), Phys. Rev. Lett. 126, 197202), or quantum-classical spin-spin models (Phys. A: Math. Theor. 56 144002).In these contexts, the Ehrenfest approximation can give a reasonable description when the trajectory is taking place close to a single potential energy surface (J. Chem. Theory Comput. 2009, 5, 728). Thus we expect overall similar scenarios in our model for the interaction between the cavity phonons and the Caldeira-Leggett's oscillators. Furthermore, since our approach to cavity leakage does not aim to a quantitative realistic description, but rather to explore qualitative trends, the lack of detailed balance should not be a crucial hampering factor. Additionally, as already said in the original version of the paper, using classical oscillators is not expensive and keeps the quantum dynamics unitary and hermitian. Due to the presence of the atomic motion, our molecule+cavity+bath is far more complex (certainly to perform numerical calculations) than, say, a spin-boson model. The use in our problem of fully quantum methods such as complex exterior scaling, Lindbladh master equations, NEGF is of course highly appealing, but these have their own (and sometimes important) limitations. We have now added a very short sentence in the paper to address this point.

………….
R3.-
As one of the main aspects of the paper is the study of fluorescence and SHG spectra on the pumping rate, the authors should show and discuss explicitly the pumping protocol that they have considered. They only mention the value of the pumping speed.

** A3 **
We thank the Referee for pointing out that the definition of the pump field is missing. In the revised manuscript we have included the coupling of the cavity field to an external pump, for which the Hamiltonian is of the form $V_{\rm ext} =g_d f(t) \sin(\omega t) (b^\dagger + b)$ where $g_d$, $f(t)$ is a pulse envelope function and $\omega$ is the frequency of the cavity field.

---

## Round 2 · List of Changes

Where applicable, at the end of each descriptor lines where in the revised manuscript the change occurs are indicated.

———GENERAL changes

  • Throughout the whole paper, we have corrected typos, and improved the choice of symbols/fonts for some quantities (for example, to avoid confusion within the atom mass M, the electronic operator \hat{M}--> \hat{\mathcal{M}}.

  • We have provided some general remarks about the interplay of exponential damping and Caldeira-Leggett baths for leakage (lines 95-105)

  • The results in Fig 3 and 4 are now obtained with either i) only exponential damping or ii) exponential damping + Caldeira-Leggett baths. We modified the description/ discussion of such figures accordingly, and we removed he original Appendix 5, where results
    from either a) exponential damping only or b) CL bath only were compared.

  • We have corrected some typos and streamlined part of App. 1 (lines 271-279)

  • We have provided additional detail about the Caldeira-Leggett model in App. 4, in the presence of the counterterm.

  • We have introduced a new Appendix (App. 5) which discusses in detail the role of the extra term in the CL model, and addresses the questionability of a CL model without counterterm.

———Changes following from REPORT 1 - We have amended some notational inconsistencies in Fig.1 and the inherent discussion as suggested in Report2, and we have readjusted locally the text in the paper to clarify the nature of the initial state in the calculations of Fig.1 (lines 163-166)

  • We have corrected the way to refer to Figs. 2 and 6, as requested in Report 1, and we have modified a sentence that now reads “..but noticeably larger for the coherent case…” (line 178)

  • We have added three references (Refs 36,51,52). Two of them (Refs 51,52) were introduced following the suggestion in Report 1.

  • We have made small changes to improve the readability and logical structure of the manuscript. We have also corrected
    notational inconsistencies at several points, for example about P(w,t) is now replaced by P(t,w) everywhere.

——— Changes following from REPORT 2 - We have included in the CL model for leakage a counterterm that ensures translational invariance, as requested in Report 2, and this is discussed in App.4 and 5.

  • The equation of the overall Hamiltonian, (Eq.1) is now numbered.

——— Changes following from REPORT 3 - We have addressed briefly the shortcoming of the Ehrenfest approximation in relation to detailed balance, when we introduce it for the treatment of the CL bath. (lines 197-202)

-We have given a detailed characterization of the external pump.This was before in a footnote, and it has now been included in the main text. (lines 110-118).

---

## Editorial Decision

published